# Unlocking history through automated virtual unfolding of sealed documents imaged by X-ray microtomography

Jana Dambrogio [1,10✉], Amanda Ghassaei [2,10✉], Daniel Starza Smith [3,10], Holly Jackson [4,10], Martin L. Demaine[5], Graham Davis [6✉], David Mills [6], Rebekah Ahrendt [7], Nadine Akkerman [8], David van der Linden [9] & Erik D. Demaine [5]

Computational flattening algorithms have been successfully applied to X-ray microtomography scans of damaged historical documents, but have so far been limited to scrolls, books, and documents with one or two folds. The challenge tackled here is to reconstruct the intricate folds, tucks, and slits of unopened letters secured shut with "letterlocking," a practice—systematized in this paper—which underpinned global communications security for centuries before modern envelopes. We present a fully automatic computational approach for reconstructing and virtually unfolding volumetric scans of a locked letter with complex internal folding, producing legible images of the letter's contents and crease pattern while preserving letterlocking evidence. We demonstrate our method on four letterpackets from Renaissance Europe, reading the contents of one unopened letter for the first time. Using the results of virtual unfolding, we situate our findings within a novel letterlocking categorization chart based on our study of 250,000 historical letters.

[1] Wunsch Conservation Laboratory, Massachusetts Institute of Technology (MIT) Libraries, Cambridge, MA, USA. [2] Adobe Research, San Francisco, CA, USA. [3] King's College London, Department of English, Virginia Woolf Building, London, UK. [4] Department of Electrical Engineering and Computer Science, Massachusetts Institute of Technology, Cambridge, MA, USA. [5] Computer Science and Artificial Intelligence Laboratory, Massachusetts Institute of Technology, Cambridge, MA, USA. [6] Queen Mary University of London, Institute of Dentistry, London, UK. [7] Utrecht University, Department of Media and Culture Studies, Utrecht, The Netherlands. [8] Leiden University, Leiden University Centre for the Arts in Society (LUCAS), Faculty of Humanities/English Literature, Leiden, The Netherlands. [9] Radboud University, Faculty of Arts, Department of History, Art History and Classics, Nijmegen, The Netherlands. [10]These authors contributed equally: Jana Dambrogio, Amanda Ghassaei, Daniel Starza Smith, Holly Jackson. ✉email: jld@mit.edu; amanda.ghassaei@cba.mit.edu; g.r.davis@qmul.ac.uk

The letter is one of the most important communication technologies in human history. Before the proliferation of mass-produced envelopes in the 1830s, most letters were sent via letterlocking, the process of folding and securing writing substrates to become their own envelopes. Letterlocking was an everyday activity for centuries, across cultures, borders, and social classes, and plays an integral role in the history of secrecy systems as the missing link between physical communications security techniques from the ancient world and modern digital cryptography[1–3]. While our attention is naturally drawn to a letter's written contents, the material evidence on surviving opened letters, such as crease marks and wax seals, testifies to thousands of folding techniques used over the centuries to turn a flat sheet of paper into a secure letterpacket. With careful study, this evidence can be used to reverse-engineer historical letterpackets, which themselves become a key dataset for the study of historical communications security methods.

Our study of 250,000 historical letters has produced the first systematization of letterlocking techniques (discussed below in "Results" section). We observe built-in tamper-evident locking mechanisms that deter potential interceptors by irreversibly damaging a letterpacket on opening, enabling the intended recipient to detect so-called "man-in-the-middle" attacks. We use these mechanisms to help assign security scores. However, since letters are designed to be opened at their intended destination, our system necessarily relies on inference drawn from surviving opened documents. A European postmaster's trunk preserving 300-year-old undelivered post, the Brienne Collection[4], provides a rare opportunity to study sealed locked letters. The trunk contains 3148 cataloged items, including 2571 opened letters, fragments, and other documents, and 577 letterpackets that have never been opened. Figure 1 shows four Brienne letterpackets studied in this paper (each measuring ~50 mm×80 mm), illustrating a variety of locking techniques despite similar exterior forms.

Until now, our analysis of letterlocking has been limited by the standard archival practice of cutting open sealed letters on request, compromising the physical integrity of the unopened letterpacket. We propose an alternate conservation approach grounded in computational analysis, where letters remain intact in their locked state, yet researchers can still read their contents and identify their letterlocking mechanisms. Drawing on high-resolution volumetric scans, produced by high-contrast time delay integration X-ray microtomography (XMT), we developed virtual unfolding to prove our letterlocking theories, and elucidate a historically vital—but long underappreciated—form of physical cryptography.

## Results

**Virtual unfolding**. Here, we present a fully automatic computational method for reconstructing and virtually unfolding letters imaged by volumetric scanning. Without any prior information about the folded shape of a letterpacket (e.g., expected number of layers and types of folds), our virtual unfolding pipeline generates: (1) a 3D reconstruction of the folded letter; (2) a corresponding 2D reconstruction representing its flat state; (3) a mapping between 3D and 2D; and flat images of both (4) the surface of the writing substrate, and (5) each letterpacket's crease pattern (Fig. 2).

Our virtual unfolding pipeline is outlined below.

1. XMT scanning produces a volumetric dataset representing material density in 3D space; inks containing higher Z elements (such as iron, copper, and mercury) result in bright (high-density) regions within the scan (Fig. 3). The raw XMT data is scaled and quantized, resulting in a volumetric array of 8-bit, greyscale voxels.

2. Segmentation identifies layers of writing substrate, separating them from each other and their surrounding environment (e.g., air). We generate feature points that lie on the medial surface of the writing substrate with subvoxel precision, drawing principally on prior work on ridge detection in 2D and 3D image data[5]. Using local geometric information, such as the 3D orientation of the writing substrate, and the first and second partial derivatives of the volumetric scan data, we estimate the thickness of writing substrate near each feature point, remove points that exceed a maximum expected thickness (indicating the presence of multiple compressed layers, adhesives, wax seals, or other scanning artifacts, Fig. 3c–h), and link the remaining feature points together to form a 3D mesh (Fig. 4e).

The meshed surface generated by segmentation typically covers ~50–80% of the complete letterpacket depending on how severely paper compaction and scanning artifacts affect layer visibility. Though the letterpackets analyzed in this study typically consist of one largely unbroken rectangular surface and (possibly) a separate paper lock, at this stage the meshed reconstruction may contain many holes and be split across several connected components. In fact, a typical letter from our dataset generates ~10,000–20,000 connected components, ranging in size from small fragments of 50 vertices to meshes of nearly 20 million vertices (Fig. 4m, n).

3. Flattening computes a distortion-minimizing 2D embedding of the segmentation result, corresponding to the document's unfolded state (Fig. 4f). Our pipeline maps mesh vertices from $\mathbb{R}^3$ to $\mathbb{R}^2$ by placing a seed triangle at the origin of $\mathbb{R}^2$ and incrementally adding vertices to this seed, subject to orientation and distance constraints. As vertices are mapped to $\mathbb{R}^2$, we reduce global edge length distortion of the 2D embedding by minimizing the energy of a system of springs. This mapping is computed separately for each connected component in $\mathbb{R}^3$, resulting in many 2D embeddings that lack a common reference frame.

4. Hybrid mesh propagation repairs discontinuities in the segmentation result on the folded and flattened embeddings simultaneously (Fig. 4g, h), and automatically merges flattened connected components into a common reference frame in $\mathbb{R}^2$ (Fig. 4m). Our hybrid approach grows the mesh into regions, where segmentation could not reliably generate feature points, while simultaneously running a joint 2D/3D optimization of vertex positions to satisfy developability, substrate layer separation constraints, and fit to the underlying XMT scan data. As separate connected components grow near each other in 3D, we compute a rigid transformation to bring their 2D embeddings into the same reference frame and successively merge many smaller connected components into a single mesh. Past work on mesh propagation in 3D scanning[6–9] aims to find aesthetically pleasing repairs, with no objective notion of correctness; instead, our methods leverage the volumetric scan data and the mapping between 3D and 2D to guide our reconstructions. The key insight that motivates our hybrid approach is that nearby mesh vertices in 3D may be far from each other in the 2D embedding (Fig. 5), so if we detect that vertices are distant in 2D, we know they should not be connected in 3D.

5. Texturing produces a 2D image of the virtually unfolded letter by mapping source voxels from the volumetric scan to their corresponding flattened destination (Fig. 4i, j). We overlay a grid of pixels (at a user-defined resolution) on the flattened mesh and compute each pixel's 3D position within the volumetric data, using the mapping from 2D to 3D. We extract a greyscale value for each pixel based on its subvoxel position in the scan data. At the current scanning resolution, our texturing results are comparable to a 668 dpi digital image. Users may offset the texture mapping along the surface normals of the folded mesh by some fraction of the paper thickness to achieve a better result for

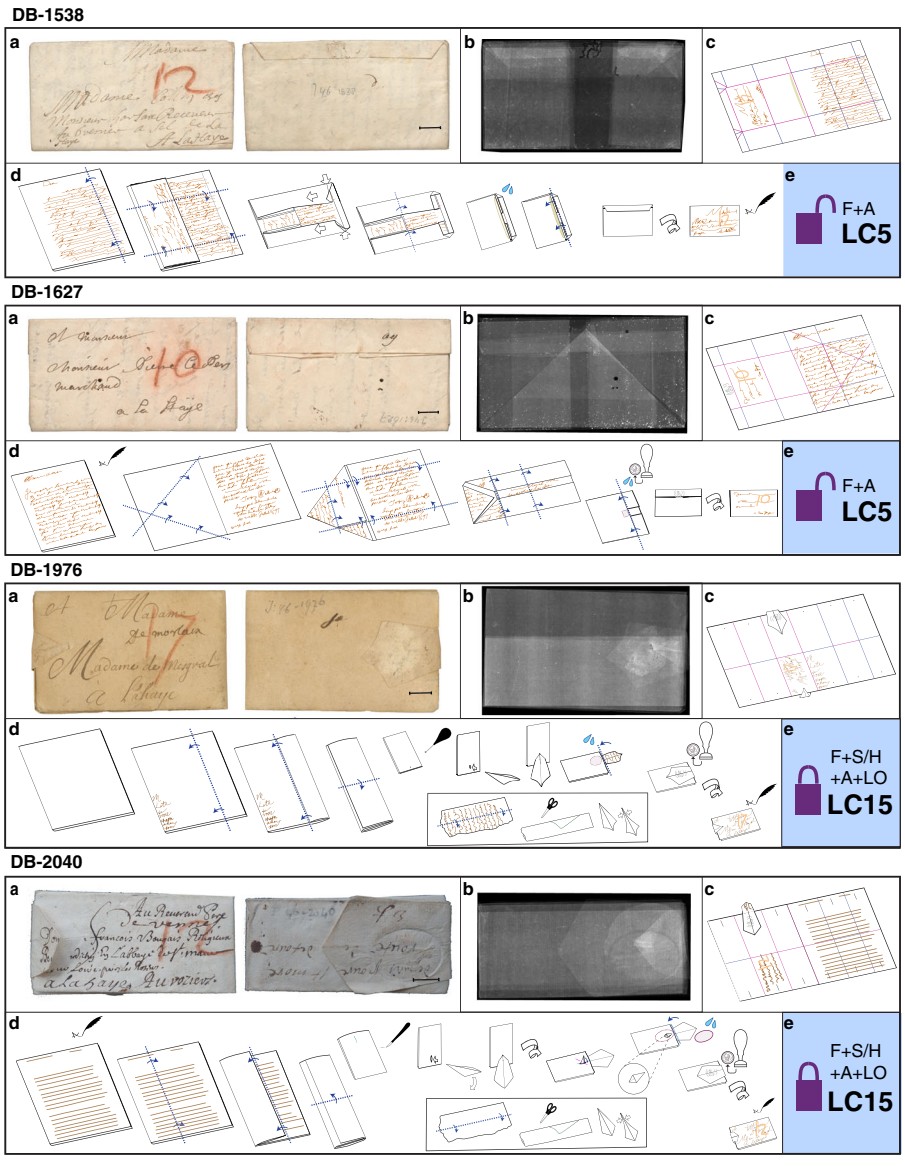

**Fig. 1 Four letterlocking examples from the Brienne Collection.** For each letterpacket (DB-1538, DB-1627, DB-1976, and DB-2040; "DB" numbers in this article refer to shelfmarks): **a** outside of letterpacket, front, and back; **b** transparent view through the volumetric XMT data showing varied distribution of layers across the letterpacket; **c** crease pattern that would be visible on the paper if fully opened, stylized from the output of virtual unfolding; **d** proposed step-by-step letterlocking process to make this packet from a flat piece of paper, based on virtual unfolding results; **e** letterlocking category (LC#) assigned based on virtual unfolding findings, letterlocking categories, and formats chart (see Fig. 7), apart from opened letter DB-2040, assigned based on surviving letterlocking evidence. For videos of the letterlocking process for each letter, see our Supplementary website brienne.org/unfolding. Scale bars: **a** 10 mm.

thicker writing substrates[10–13]; however, we found zero offset gives the best results for our dataset.

A similar process is used to generate an image of the crease pattern; in this case, we map the mean curvature of the 3D mesh to 2D (Fig. 4k, l). The sign of the curvature corresponds to the direction (mountain or valley) of the crease, indicated by red or blue, respectively. Color saturation specifies the magnitude of curvature, with white indicating zero curvature. The resulting crease pattern images show sharper creases as thin lines of saturated color, and more gentle creases as broad, faded lines. In addition to the folding pattern, other features that introduce curvature into the writing substrate—such as buckling near the corners of a letterpacket—are made visible (Fig. 2f). In combination with the 3D reconstructions, these crease patterns enable researchers to recreate the step-by-step locking process that would be needed to turn a flat sheet of paper into the finished packet (Fig. 1d).

**Reading the unread.** One of our major results is to reveal for the first time the contents of DB-1627, a letter typical of day-to-day communications of the time. It contains a request from Jacques Sennacques, dated July 31, 1697, to his cousin Pierre Le Pers, a French merchant in The Hague, for a certified copy of a death notice of one Daniel Le Pers (see Supplementary Note 1 for images, transcripts, and translations). Before computational analysis, we only knew the name of the intended recipient, written on the outside of the letterpacket.

**Virtual unfolding.** Virtual unfolding produced nearly complete reconstructions of unopened letterpackets DB-1538 (Fig. 2b–f), DB-1627 (Fig. 6a, b), DB-1976 (see Supplementary Notes 1 and 3), and the opened test-case DB-2040 (partially shown in Fig. 6c), illustrating the generality of our approach. For letters with a separate paper lock (DB-1976 and DB-2040), we are able to reconstruct the

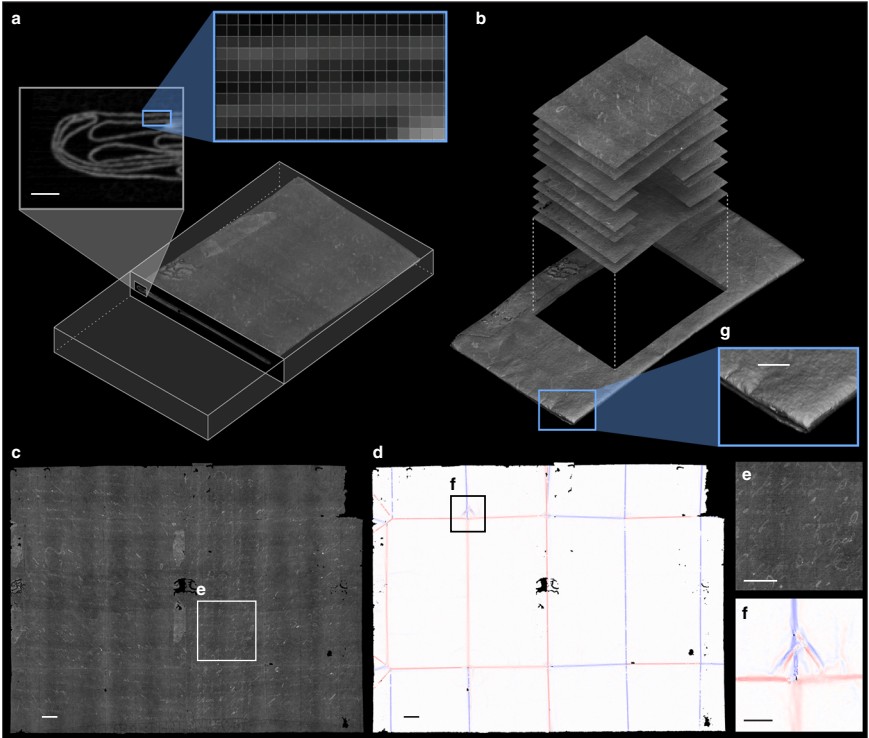

**Fig. 2 Virtual unfolding overview.** High-level overview of virtual unfolding showing. **a** XMT volumetric data and resulting **b** folded and **c**, **d** unfolded reconstructions of unopened letter DB-1538. **c** XMT scan data is mapped to 2D to produce an image of the sealed letterpacket's contents. **d** Curvature is mapped to 2D to generate an image representing the crease pattern of the folded letterpacket. **e** Ink visible on the surface of the paper shows the presence of double-sided text (running both vertically and horizontally). **f** A close-up of the crease pattern image depicts sharp creases on the interior of the packet as thin, saturated lines, and rounded creases toward the outside of the packet as diffuse lines, due to their lower curvature spread across a larger region. Surface curvature induced by buckling of the paper near the corner of the letterpacket is also revealed by this analysis (also visible in the 3D reconstruction, **f**). Scale bars: **a** 1 mm, **c**–**e** 10 mm, **f**, **g** 5 mm.

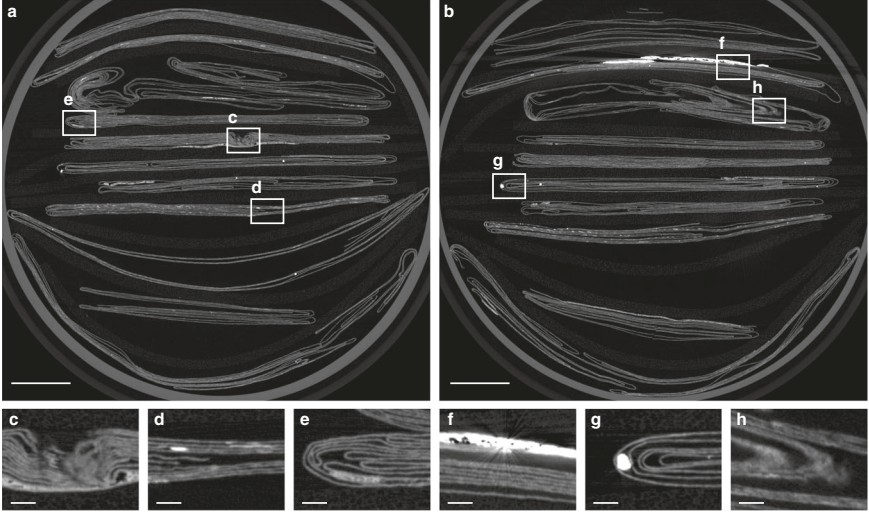

**Fig. 3 XMT scans of 11 letterpackets.** Two sample cross sections of XMT data at **a** $z = 525$ and **b** $z = 1611$, showing 11 letterpackets. From top to bottom, letter IDs are DB-1465, DB-1818, DB-2004, DB-1538, DB-1976, DB-1627, DB-2199, DB-2156, DB-1888, DB-2040, and DB-1876. Note that DB-1888 is not visible in cross section **b**. **c**–**h** Segmentation challenges associated with XMT data. **c** Paper layer compaction results in ambiguous boundaries between layers. Dense regions of **d** ink, **e** adhesive, **f** sealing wax, and **g** pounce sand (used to absorb ink) interfere with neighboring regions of scan. **h** Paper may be angled relative to a given cross-sectional plane, resulting in a blurred, artificially thickened appearance in a 2D cross section. Scale bars: **a**, **b** 10 mm, **c**–**h** 1 mm.

primary writing substrate and the paper lock as two separate pieces. We validated our results by comparing the reconstruction of the opened letter (DB-2040) to a photograph of its contents, showing low geometric distortion (Fig. 6c, d). Additional information about our methods can be found in Supplementary Methods.

See Supplementary website brienne.org/unfolding for additional images, animations, and foldable models; a selection of these images has been included in Supplementary Information, including an animation showing a physics-based unfolding simulation of the final reconstruction of DB-1538 (Supplementary Movie 1).

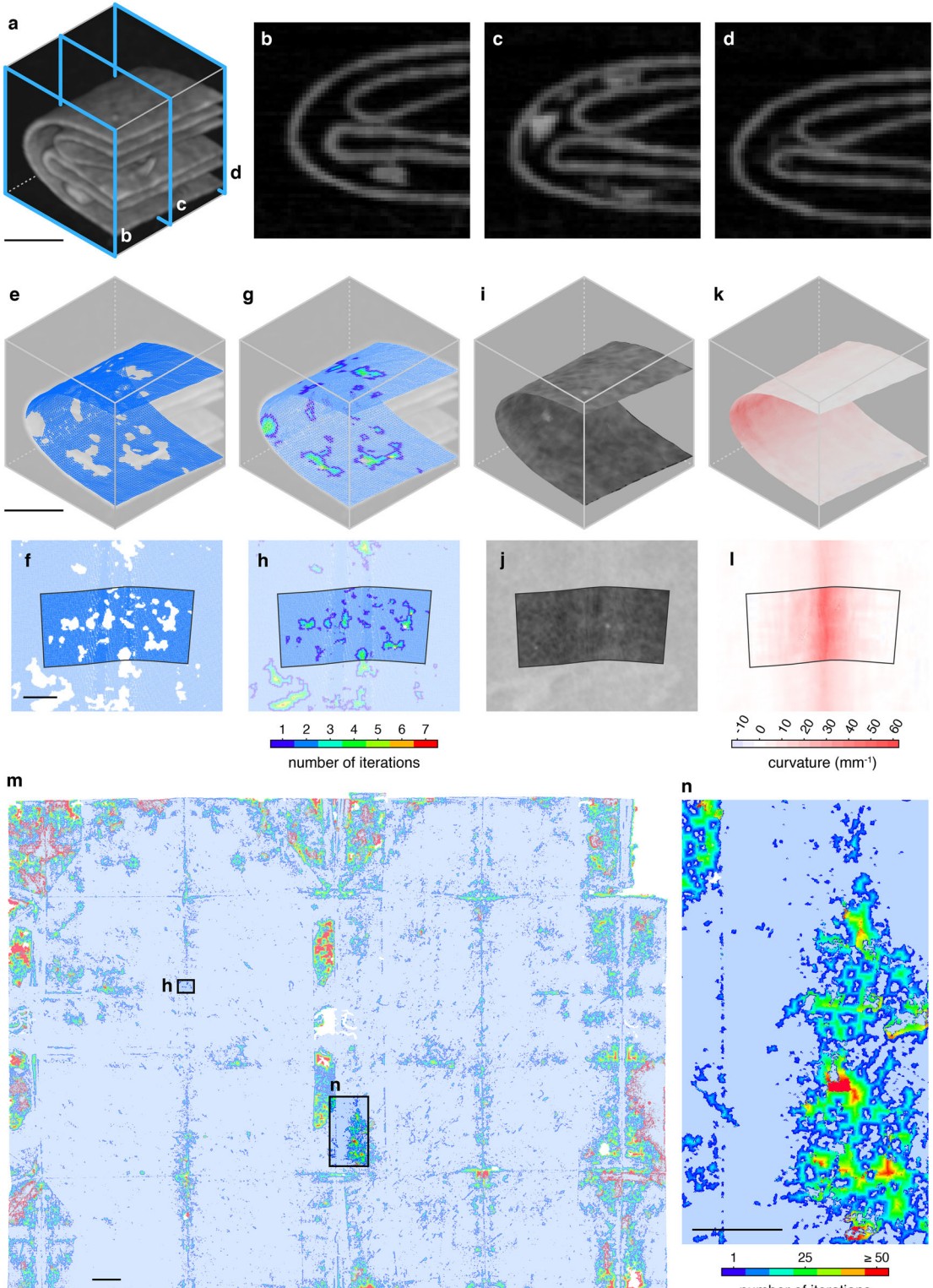

**Fig. 4 Virtual unfolding pipeline. a** Cropped 3D volume of DB-1538 XMT data showing three sections of folded paper; **b-d** three cross-sectional slices of the volume; **e** 3D segmentation result for outermost paper section; and **f** corresponding flattened state. **g** 3D mesh after hybrid mesh propagation, with color indicating the order that new vertices were added to fill in segmentation holes, and **h** corresponding flattened state. 3D mesh reconstruction colored with texturing results showing **i** greyscale XMT data image and **k** crease pattern image, and corresponding flattened states (**j**, **l**). Black boundary in **f**, **h**, **j**, **l** shows the bounds of the 3D crop mapped to 2D. **m** Final flattened reconstruction showing the initial segmentation result in pale blue with hybrid mesh propagation additions shown in a spectrum of colors according to the order in which new vertices were added. **n** A close-up reveals numerous isolated connected components in pale blue that were merged to form one continuous mesh during this process. Scale bars: **a**, **e**, **f** 1 mm, **m** 10 mm, **n** 5 mm.

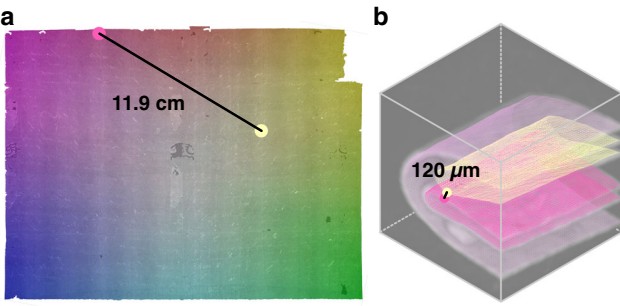

**Fig. 5 Distance map applied to 2D and 3D embeddings.** Colored 2D distance map of the **a** flattened reconstruction of DB-1538 (same as Fig. 2c), and the same colormap applied to **b** a cropped section of the 3D folded state demonstrates that nearby vertices in 3D may be very far apart from each other in 2D, if they belong to different layers of the writing substrate. Distance metrics from the 2D and 3D embeddings of the mesh are used during hybrid mesh propagation to enforce global developability of the final result.

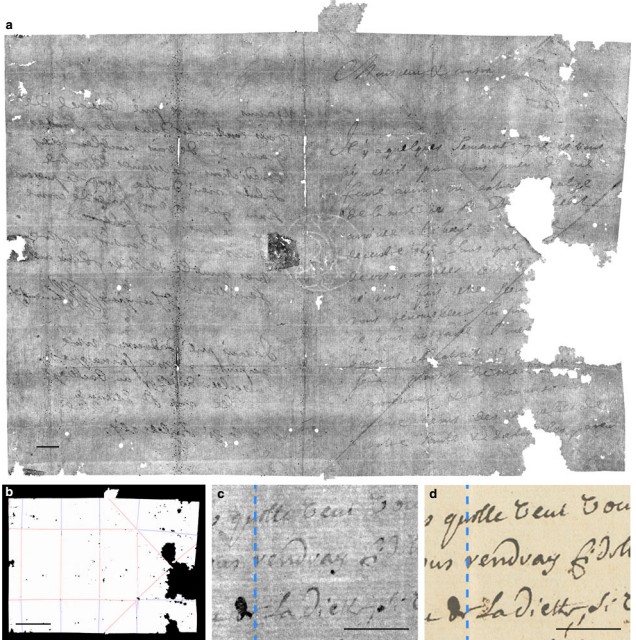

**Fig. 6 Virtual unfolding results.** Virtual unfolding results for unopened letter DB-1627 (**a**, **b**) and opened letter DB-2040, showing **a** XMT-textured image after virtual unfolding (this image has been inverted and linearly scaled to increase contrast of text and other details, such as a visible watermark in the center and laid and chain lines of the handmade paper); **b** crease pattern image; and **c** a close-up view of a portion of the final texturing results for opened letter DB-2040 (inverted and linearly scaled), compared to **d** a photograph demonstrates low geometric distortion of the virtually flattened result across a crease (indicated by the dashed blue line). Scale bars: **a** 10 mm, **b** 50 mm, **c**, **d** 10 mm.

**Letterlocking categories and formats.** Virtual unfolding confirms the categorization system of letterlocking, without perturbing the rare historical artifacts studied (Fig. 7). Until now, the categorization system of letterlocking was defined by making simulacra (counterpart models) of opened historical letters in the presence of the originals. Virtual unfolding enables the analysis of authentic, closed examples to accurately discern their undisturbed engineering mechanics. We characterize letterlocking mechanisms according to their manipulations: folds/rolls, tucks, slits/holes, adhesive, and/or locks; we further subdivide locks into

three types. Together, these manipulations indicate letterlocking "category" and level of security. We define the visual "format" of closed letterpackets according to their silhouette and orientation relative to the address (for letterlocking terminology, see Supplementary Note 2). Some document manipulations have been observed and described before[2,14–17], but our study is the first to propose a complete systematization of evidence and tie it consistently to security.

As Table 1 shows, the Brienne Collection contains 3148 items: 1706 opened items we count as letterlocking (this number includes 5 partially opened items); 865 opened items we do not count as letterlocking; plus 577 unopened letterpackets; these unopened letterpackets may themselves contain further items, meaning that we do not yet know the total number of documents in this collection. The 1706 items counted as opened locked letters were identified, using the letterlocking categories and formats chart (Fig. 7). Table 2 demonstrates that, among these items, we count seven formats, mostly rectangles but also featuring ten squares, three square diamonds, and a pentagon. We also place 1475 of these opened letters into 11 categories, with LC5 and LC6 clearly the most numerous. A further 231 opened locked letters could be one of two categories, and two other opened letters could fall into three categories; because they have been opened, we can no longer tell if these letters used the tuck manipulation, which would change their category. Keeping the remaining 577 letters unopened and using virtual unfolding allows us to assign their category with complete confidence. Examined from the outside only, unopened letter DB-1976 might have been an LC15 low security or LC23 high security, but our virtual unfolding result determines that it is LC15 low security.

## Discussion
Our virtual unfolding pipeline builds on prior work using XMT to extract text hidden inside sealed, damaged, or otherwise unopenable historical documents[18]. Recent investigations lay out a computational framework for virtually unwrapping or unrolling these documents to a flat isometry so their contents can be more readily observed; to date these investigations have focused primarily on scrolls[10–12,19–27], books[13,28,29], and artifacts folded once or twice[28,30,31]. The primary motivation for many prior studies has been to recover text from highly damaged documents that cannot be physically opened and read. Damaged artifacts and highly fibrous, delaminated, or otherwise heterogeneous writing substrates pose particular problems for segmentation, and severe damage-induced warping may be difficult to virtually flatten without introducing heavy distortion of the surface. Because the artifacts we study are made from a largely homogenous paper material that is relatively undamaged, our focus instead has been to fully automate and extend existing computational flattening frameworks to tackle a new class of documents with more intricate folding than has previously been explored.

Through the four examples studied here, we demonstrate the generality of our approach. Each letterpacket was constructed with a distinct folding sequence: some packets contain inner folds that are angled diagonally relative to their outer silhouette; others are bound together by a separate paper lock. Rather than linking together results from many independent cross-sectional analyses of the volumetric data (assuming some degree of global layer congruence across the length of the scan[10–12,19,20,22–24,26,30,31]), we utilize a fully 3D geometric analysis that is unbiased to fold orientation and makes no assumptions about the folded topology of the packet. In addition, our methods do not require user-defined contour lines for initialization[11,20,24,25,30,31] and are scalable to writing substrates of different thicknesses. Our results suggest that our approach may apply broadly to many types of

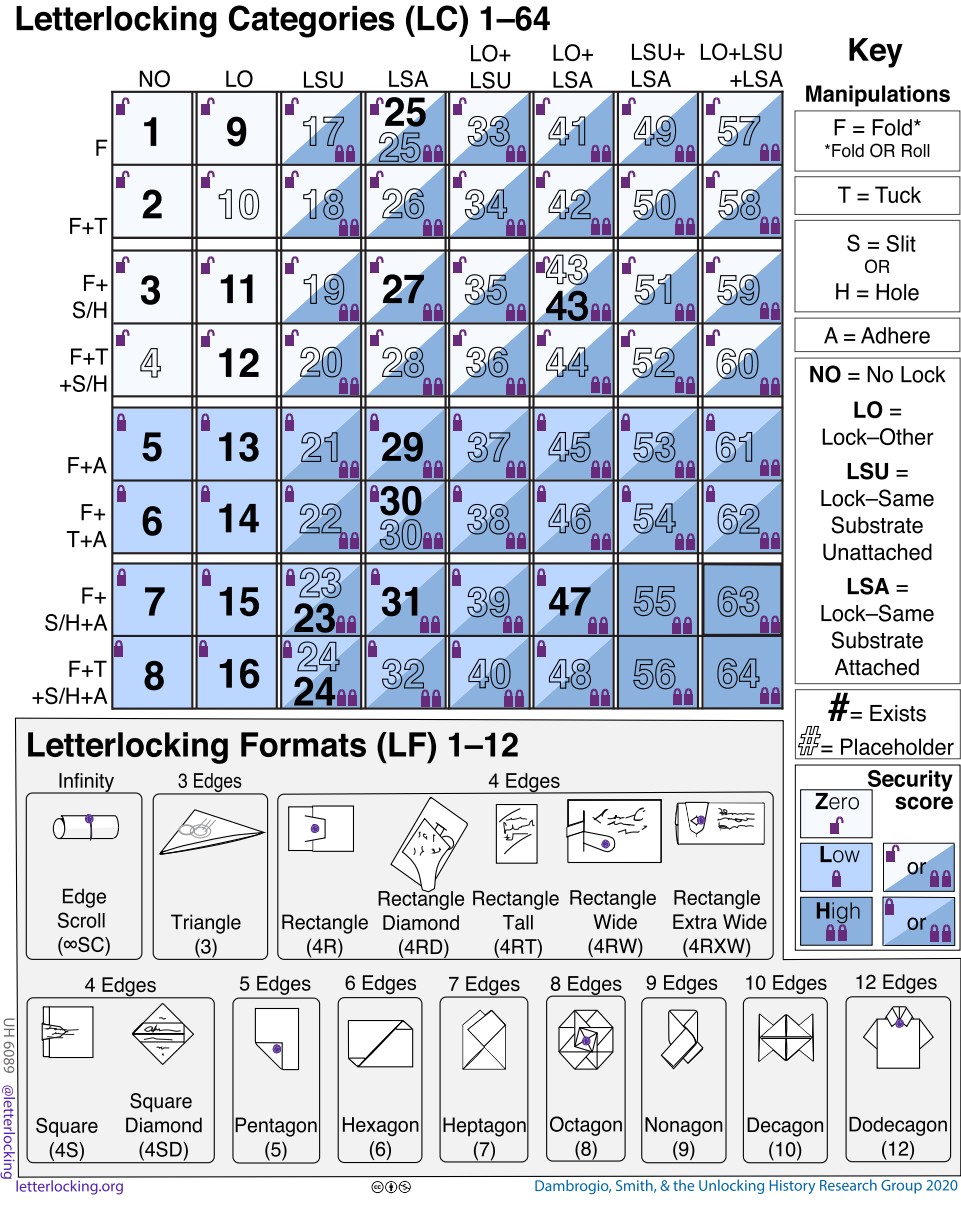

**Fig. 7 Letterlocking categories and formats chart.** Letterlocking categories 1–64, based on manipulations and assigned security score. Filled letterlocking category numbers indicate that historic originals exist; unfilled numbers indicate hypothetical categories. Letterlocking formats show up to 12 edges with indicative examples.

| Table 1 Breakdown of Brienne Collection by number. | | | | |
|---|---|---|---|---|
| Total items | Items that are not counted as letterlocking | Unopened items | Opened items counted as letterlocking | Opened items in one or more categories |
| 3148 | 865 | 577 | 1706 total 1701 opened 5 partially opened | 1706 total 1475 assigned to one category 231 either-ors |

historical texts (including letters, scrolls, and books), since our method can handle flat, curved, and sharply folded regions alike. Other potential applications include analyses of developable surfaces, such as origami and sheet materials (e.g., plastics and metals) used in manufacturing.

To facilitate widespread adoption of virtual unfolding, future work could include improving robustness. Our methods rely heavily on local interactions between vertices in the mesh; while efficient to compute, these local rules can be prone to global

consistency errors. This is exacerbated by the presence of scanning artifacts (Fig. 3), especially those caused by leaded seals (i.e., those containing red lead oxide) or shellac seals containing strands of metal, which were avoided in this study (the seal shown in Fig. 3f is unleaded). Errors in segmentation and flattening may accumulate into localized distortions of the 2D embedding and result in incomplete or incorrect merging of connected components during hybrid mesh propagation; the missing pieces of letters DB-1538 and DB-1627 are a result of these merging issues

**Table 2 Breakdown of Brienne Collection by letterlocking format (A) and category (B and C).**

| A. Letterlocking format (LF) | Count |
|---|---|
| 4R | 58 |
| 4RW | 1473 |
| 4RXW | 139 |
| 4RT | 22 |
| 4S | 10 |
| 4SD | 3 |
| 5P | 1 |
| Total | 1706 |

| B. Letterlocking category (LC) | Security score | Count |
|---|---|---|
| LC1 | Zero | 41 |
| LC2 | Zero | 3 |
| LC5 | Low | 760 |
| LC6 | Low | 612 |
| LC13 | Low | 1 |
| LC14 | Low | 1 |
| LC15 | Low | 6 |
| LC16 | Low | 3 |
| LC25 | Zero | 1 |
| LC29 | Low | 46 |
| LC30 | Low | 1 |
| Total | | 1475 |

| C. Letterlocking category either-ors | Security score | Count |
|---|---|---|
| LC1 or LC2 | Zero | 41 |
| LC1 or LC5 | Zero or low | 1 |
| LC5 or LC13 | Low | 1 |
| LC5 or LC29 | Low | 4 |
| LC5 or LC6 | Low | 180 |
| LC5 or LC6 or LC29 | Low | 2 |
| LC6 or LC29 | Low | 1 |
| LC29 or LC30 | Low | 1 |
| Total | | 231 |

(Fig. 8). Future work may include additional passes to fill in the gaps in the reconstruction or user-in-the-loop mechanisms to resolve segmentation errors early in the process.

Inks with a similar density as the underlying writing substrate (e.g., carbon-based inks) may have low contrast (Fig. 2c, e) or no discernible signal at all (see DB-1976 result in Supplementary Notes 1 and 3). Ink contrast and legibility could be improved by using alternative scanning methods[32] or by performing additional post-processing on the unfolded results to amplify small changes in density or structure from the presence of ink[33].

In general, all files created from our pipeline are derivations of the original scan data and code; given changes to either of these inputs, we would expect the results to change. As improvements are made to the pipeline, it is crucial to track metadata, such as the code version and changes to input parameters to ensure reproducibility of results. In addition, tools to visualize the transformation of data through the pipeline will give researchers a window into the data provenance, affirming the integrity of the results, especially in cases where the original physical artifact remains unopened.

A potential obstacle to applying this pipeline to highly warped or damaged documents is our incremental flattening approach, which maps the results of segmentation to $\mathbb{R}^2$. Because our datasets have so far been limited to artifacts with little warping, we do not have to introduce much distortion into the mesh to compute this mapping (Fig. 8), and our incremental approach is usually sufficient. Many historical artifacts are warped due to excessive heating, water damage, delamination, decay, or other processes of time[34], and may fail to map properly to 2D in our current scheme. An extensive body of research from computer graphics solves the problem of computing a globally consistent 2D embedding of a 3D triangle mesh for the purpose of texture mapping, and much of this work focuses on distortion minimization[35–37]. The main technical challenge in applying this work to our current pipeline is that it generally assumes the 3D mesh is topologically equivalent to a disk (i.e., has no holes), so we leave these engineering challenges for future work.

Our work seeks to make an intervention in the conservation of cultural heritage. Once a document such as an unopened letter is damaged in the opening process, we lose a sense of the object as untouched and intact. The material evidence that a letter preserves about its internal security—including highly ephemeral evidence about tucks and layer order, which usually leave no material trace—can now be retained for investigation. Our methods therefore create an opportunity for the heritage sector to protect the integrity of documents even where there is a need to access their contents.

Both opened and unopened letters can now be investigated for their letterlocking evidence, establishing the methodological basis for a new discipline. The category and format chart (Fig. 7) establishes a consistent set of terms and methods designed so they can be incorporated as metadata in the catalogs of archives and libraries worldwide, in order to enable the study of letterlocking as a global phenomenon. Letterlocking metadata has already been trialed successfully alongside content details in Early Modern Letters Online (EMLO), a union catalog of letters from the Renaissance world[38]. Data-driven work already yields interesting results within the Brienne Collection: the dominance of LC5 and LC6 categories shows a marked shift in letterlocking usage, away from the "fold, tuck, and adhere" method so common earlier in the century, and towards the "fold and adhere," often producing packets that look much more like the modern envelope. As well as confirming the conclusions of a two-decade letterlocking study, virtual unfolding promises to accelerate future research. It took a decade of research to conclude that the last letter of Mary, Queen of Scots (National Library of Scotland, Adv. MS. 54.1.1)—a document of great national importance in Scotland—employed a highly secure spiral locking mechanism[39]. Virtual unfolding could produce a result in a matter of days.

Our research has the potential to unlock new historical evidence, starting with the Brienne Collection. Although DB-1627's written contents might not in themselves radically change our understanding of 17th-century history, as further text from the Brienne Collection becomes available through virtual unfolding, it will be analyzed alongside already-open letters studied by historians on the Signed, Sealed, and Undelivered project[40]. New Brienne Collection data may contribute to broader studies of politics, religion, migration, music, drama, and postal networks in early modern Europe.

Virtual unfolding stands to make an impact far beyond this trunk of correspondence, too, since collections worldwide contain unopened letters and documents, from many places, cultures, and historical periods. One important example is the hundreds of unopened items among the 160,000 undelivered letters in the Prize Papers, an archive of documents confiscated by the British from enemy ships between the 17th and 19th centuries[41,42]. If these can be read without physically opening them, much rare letterlocking data can be preserved.

The procedures outlined here advance an emerging conceptual shift in the digital humanities by adding to the body of methods that cross the digital–material divide. Conserving intact these records of human interaction with materials, while making their secrets visible, enables a perspective on history that is both kinetic and tactile, and which encourages new ways of thinking about the lives, emotions, and creativity of historical individuals and communities. Doing so also challenges cultural historians to

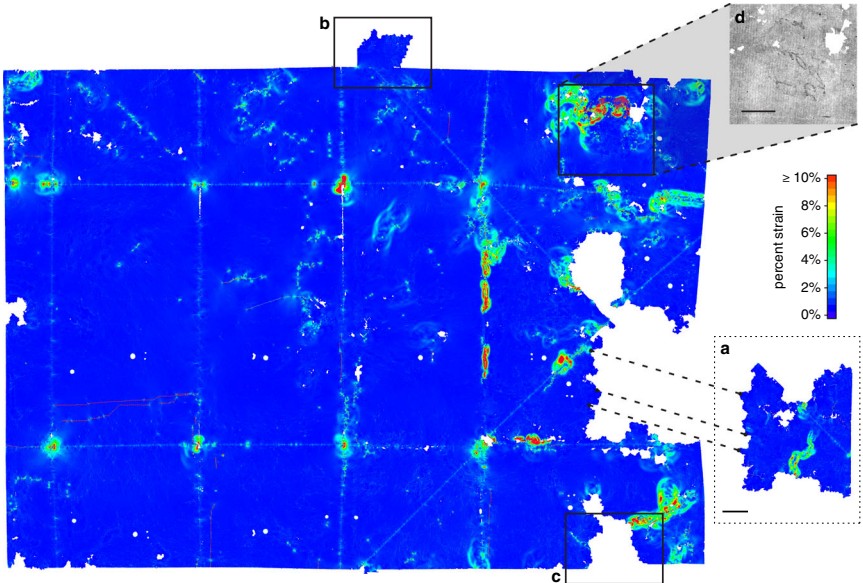

**Fig. 8 Strain analysis of 2D embedding.** Edge strain map (measured as edge length deviation in the 2D embedding as a percentage of nominal length in 3D embedding) for DB-1627 shows a nearly isometric mapping of the 3D embedding to a plane. Localized strain concentrated around creases (especially where creases intersect each other) and in other areas where poor initial segmentation leads to an accumulation of error during hybrid mesh propagation. Distortions of the 2D embedding can negatively impact automatic merging of connected components as demonstrated by: the large region **a** that was not successfully merged to the final result; the region **b** that was incorrectly positioned in the final result (should have been located at **c**); and the region **d** containing text that was incorrectly positioned in the final result. Scale bars: **a**, **d** 10 mm.

reconceptualize hidden, secret, and inaccessible materials as sites of critical inquiry; letterlocking and virtual unfolding point to the ways that history sometimes resists scrutiny, and that resistance itself deserves patient study.

Letterlocking has enormous potential to cast new light on countless primary historical materials, and our generalized algorithmic conservation approach demonstrates the power of computational analysis for driving this research forward. We envision a thorough, data-driven study, encompassing tens of thousands of known unopened letters plus millions more opened letters, drawing together letterlocking data globally to make persuasive, consequential statements about historical epistolary security trends. By synthesizing traditional and computational conservation techniques, we can help further integrate computational tools into conservation and the humanities—and show that letters are all the more revealing when left unopened.

## Data availability

The original XMT datasets, results generated by the virtual unfolding pipeline, and datasets supporting letterlocking findings are available on Dataverse (https://dataverse.harvard.edu/dataverse/uharticle) and https://doi.org/10.7910/DVN/H2NUGV. The documentation of the research process for letterlocking has been archived at Massachusetts Institute of Technology Libraries, Department of Distinctive Collections, as the Unlocking History Research Group Collections (MC0760).

## Code availability

Our code is publicly available on Github (https://github.com/UnlockingHistory/virtual-unfolding/), with an archived version at https://doi.org/10.7910/DVN/VBWOI6. For further details on the processing pipeline see Supplementary Methods.

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

## Acknowledgements

We are grateful to Sound and Vision The Hague (Beeld en Geluid Den Haag), especially curators Koos Havelaar and Ruben Verwaal for making Brienne Collection materials available, and to Anja Tollenaar and Sarah Jane Earle for facilitating our work. The Unlocking History research group has helped develop the technical language and resources behind the study of letterlocking, particularly Ayako Letizia, Emily Hishta Cohen, Laura Bergemann, Brien Beidler, Jennifer Pellecchia, Mary Uthuppuru, and Tomas Brown. Chris Bourg at MIT Libraries, Kenny Cheung at NASA, and Neil Gershenfeld at MIT have supported this project from the beginning. Dennis Flynn, Barbara Hosein, Mike Tarkanian, and Henry Woudhuysen gave us important feedback about letterlocking. At the University of Oxford, Cultures of Knowledge and Early Modern Letters Online supplied us with meeting space at a crucial point in the research, with special thanks to Miranda Lewis. Letterpacket images in Fig. 1 courtesy of Sound and Vision The Hague, Fig. 1c, d, designed by J.D., Annie Dunn, and Nicole Araya; Figs. 1e and 7 designed by J.D., D.S.S., Annie Dunn, and Matthew Li. R.A., N.A., J.D., D.S.S., and D.vdL. acknowledge funding for the Signed, Sealed, and Undelivered project from an Internationalization in the Humanities Grant from the Nederlandse Organisatie voor Wetenschappelijk Onderzoek (Dutch Research Council; project code 236-69-010); Metamorfoze; and Sound and Vision The Hague. J.D. acknowledges funding from The Seaver Institute; MIT Libraries; the MIT Undergraduate Research Opportunity Program (UROP); The Gladys Krieble Delmas Foundation; Bodleian Libraries, University of Oxford; with thanks to Kaija Langley, Mary Hurley, Tess Olson, Emilie Hardman, and MIT Office of Foundation Relations. A.G. acknowledges funding from Adobe Research. D.S.S. acknowledges funding from The British Academy; The John Fell Fund at Oxford University Press; The Zilkha Fund at Lincoln College, University of Oxford; and three sources at King's College London: the English Department; the International Collaboration Fund, Faculty of Arts and Humanities; and the King's Undergraduate Research Fellow (KURF) scheme. A.G. and H.J. acknowledge support from sponsors of the MIT Center for Bits and Atoms and MIT Computer Science and Artificial Intelligence Laboratory. Any opinions, findings, and conclusions or recommendations expressed in this material are those of the authors and do not necessarily reflect the views of our funders.

## Author contributions

J.D. and A.G. should be considered joint first authors, D.S.S. and H.J. joint second authors, and all contributed equally to the writing and editing of the article. J.D. conceptualized the original research problem, assembled the team, and managed the project. J.D. and D.S.S. co-directed the Unlocking History research group and developed letterlocking research. A.G. and H.J. developed the virtual unfolding algorithm. E.D. and M.D. gave high-level advice on research question design and development, helping to finalize the article. G.D. and D.M. supervised XMT scanning and delivery of scan data to MIT. A.G., H.J., E.D., G.D., and D.M. were in charge of data curation. R.A., N.A., J.D., D.S.S., and D.vdL. co-directed the Signed, Sealed, and Undelivered project on the Brienne Collection, and together selected letters for scanning. R.A., N.A., and D.vdL. transcribed, edited, and translated the unfolded letter text, researching the historical context of the Brienne letters. All authors contributed to funding acquisition.

## Competing interests

The authors declare no competing interests.
