## [Peer Review File · Nature Communications]

REVIEWER COMMENTS

Reviewer #1 (Remarks to the Author):

In this work the authors use micro-computed tomography to acquire a volumetric scan of a folded letter. The letter may have been folded in a complex way.

Through a series of software-implemented steps, the acquired volume is converted into a single image of all the parts of the letter as if it had been opened and photographed. But since this is done from the micro-CT alone, the work is titled by the authors "virtual unfolding."

This work follows the same vein as a long line of work going back to the 1970s where medical CT, shortly after its wide deployment for medical imaging, was applied to mummies and then to fossils. Some more recent prior work, which the authors reference, applied the idea of the volumetric scanning and then analysis of the resulting virtualized data to cuneiform tablets and - separately - a lead scroll. I would like to point out that the authors should consider citing as well the work on manuscript fragments within book bindings, which was the first application similar to this work on letters: the process was defined, tomography was the basis, and the material was not morphological like cuneiform or incised surfaces but rather was ink-based:

W. B. Seales, Y. Lin, Digital restoration using volumetric scanning,
in Proceedings of the 2004 Joint ACM/IEEE Conference on Digital Libraries (IEEE, 2004)

Y. Lin, Physically-Based Digital Restoration Using Volumetric Scanning

(University of Kentucky, Lexington, 2007) Ph.D. thesis.

Y. Lin, W. B. Seales, Opaque document imaging: Building images of inaccessible texts, in Tenth IEEE International Conference on Computer Vision (ICCV'05) (IEEE, 2005)

The authors show a complete pipeline for producing a virtually-unfolded letter that is demonstrably readable. They elaborate a method for solving the problem of unfolding that they claim is fully automated. And then finally they provide a catalog of various types of folded letters from the era, with the possibility to take the data from a letter that was folded in an unknown way and then classify it into the taxonomy.

While the concept of the virtual analysis of written material is not novel ("virtual unwrapping", "virtual unrolling", "virtual unfolding" - I prefer unwrapping even if folding or rolling was involved), some of the techniques that the authors have originated to meet the challenge are new and effective. In particular, some of the optimization methods used for refining the localization of surfaces within the volume, extending their reach within the volume to improve coverage and remove holes, pruning outliers, and measuring error in developability for the ultimate step of enforcing an isometry to a planar output are interesting and valuable.

I would like to offer some thoughts that might be useful to the authors for refining the paper's presentation for the purpose of improving it.

I think the authors may be inadvertently minimizing the complexity of the objects others have approached with this general framework when they say in the abstract and then the introduction "but these methods have been limited to documents with simple folding or rolling." The fact is that the letters in this paper appear quite healthy from a conservation standpoint. The paper is not delaminated, burned, stuck together, crushed, soaked to the point of fibrous deterioration, etc. Some might argue (I am not) that the problem of virtually opening these letters can be reduced to a set of linear layers that are connected through folds that, once identified, can allow linear sections to be re-assembled without the need for extra fretting over folds, interlocking sections, etc.

I would simply caution against calling rolled material, or layers within a book or codex, as "simple." Most of it that I have seen, because of damage, is riddled with uncertainty as to what is folded, broken, crushed, full of holes, bubbled because of burns, etc.

So perhaps making it clear that your challenge is shape / category of interlocking letter, not damage. Others deal with damage, and it has its own challenges.

Also I appreciate the desire to highlight the part of this work that should be seen as novel, but I would caution against overreach with which part of these methods are new versus which parts you have appropriated - perhaps in a different/novel way for the particulars of this kind of letter/material.

General virtual unwrapping has been postulated for a while (based on any kind of

volumetric imaging) followed by software steps (in abstraction, such as "segmentation" and "texturing" and "unwrapping". Your work fits nicely as a variation into those already-articulated frameworks, and you could actually make that alignment if you chose to...to argue that you are refining / innovating a new "unfolding" step in the generally accepted pipeline of virtual analysis of written material.

Another point I believe could be made more crisp is the reality of physical opening. Is it not possible to open these letters at all through paper conservation? Would doing that ruin something? I realize that when the seal is intact, it would need to be broken - but my understanding from the paper was that the ones that were scanned did not have seals (because the lead would cause artifacts in the scan). So how do you argue against someone who might say: "keep a few exemplars intact, but open the others as a physical conservation project for the purpose of better imaging of the text."

The argument I have used is that in some cases physical restoration is impossible, e.g., it will destroy the object. But I don't think you can say that here: which part of this letter object is destroyed if you open it?

You claim that these segmentation, unfolding, and texturing methods are fully automatic ("Virtual Unfolding", page 2). That is important and I think can and should be made clearer (and more believable) if you focus on explaining it a little better. My experience is that "fully automatic" is always defeated by the condition of certain items (pages too close together; not enough spatial resolution in the volume; ink variations; etc.) It might be

instructive for you to show examples where your method fails in order to give the reader insight into the failure modes and the visual consequence in the final unfolded image of full automation when things go wrong.

You do make the statement in talking about prior work as distinguished from yours that "nearly all methods are 2.5 D (and not fully 3D)." True that many have approached the problem from a slice-based position, but not everyone. Our own work uses a 3D structure tensor in the volume and other fully-3D approaches - I am sure we are not the only ones.

I would like to point out also that the substrate of paper you are viewing in this problem - which is not burned or delaminated or apparently damaged in any way - is really the best case for segmentation. Likewise, for texturing, the ink gives a nice density-based signal, obviating the need to do any processing at the texturing step other than to situate what is in the volume onto the segmented surfaces in an accurate way. That is fine for this problem - just making the point that it is possible with different materials to face more complicated issues at that step in the processing.

It wasn't straightforward for me to understand how the unfolding algorithm detects automatically where the folds are...it seems like the mapping from e_3 to e_2 has to happen piecewise for each of many "connected components", and then those individual components, after "mesh propagation", end up being put into a common frame by solving for a rotation/translation, and then whatever *that* turns out to be helps you infer that a "fold" or "hinge" is happening...But it is hard to

tell if I'm right about this - if anything is ever explicitly identified as a fold or a crease or crack, or how that stage works in the end. [aside: I see that you are nudging the texture with curvature info to build a visual cue around "crease", but that's not what I mean].

Maybe a high level walk-through first could help the discussion.

The figure 2 really has too much in it (and it is too tiny) and I think it would be helpful to expand that if there is space.

In the end I am left wondering about how much overall distortion there is in the flattened and unfolded letter pages. Since they are undamaged and folding is non-elastic, they ought to map quite readily isomorphically to a plane. I would guess the layers are all essentially flat, meaning there isn't much digital flattening that needs to happen during the mesh optimization. Is that true?

Would it be possible to show distortion maps / heat maps from the 3D segmentation to the 2D flattened image to know where there was page distortion (geometric) from the digital unfolding step? I don't think I mean a "distance map" here, but a visual measure of the distortion on a section that had to occur to go from E3 to E2.

In the supplementary description the domain experts compare the image to the quality of micro-film. I checked a few numbers: at 38 micron voxel resolution you should be able to get somewhere around 660dpi on the final image (subject to stretching, etc.: this is just $25400 / \text{"voxel's resolution in microns"}$).

So you could get 1200dpi if you scanned at 20 microns. So that's one point - of course, maybe that isn't possible with the kit you used. A related point

is that a 40MP digital camera will give you 1000-1200 dpi on a letter sized page, and that is considered archival quality. So if you could get just a bit better at the scan stage you would be there with your output final textures. I wonder if your domain experts would find that to be sharper, more like photography?

Final though is one of provenance chains. By that I mean your final texture/image claims are derived directly from the original scan through a series of complex algorithmic operations you define. People working in this field should consider that extracted information from the interior of artifacts that can never be opened for inspection are hard to subject to peer scrutiny. There is no way as a reviewer of your work that I can truly verify your final texture claims - in particular, the textual implications of what you claim to have revealed from within these letters. You might acknowledge that somehow, and suggest some way to build confidence in those of us who must trust that the revealed text is authentically supported by the physical object and the volumetric scan you made. I consider it an open problem as yet, but if I grabbed your volume and pushed data the other way through the pipeline we could create/change text rather than reveal it...creating a pathway for inscrutable results that will not help the cause of non-invasive analysis.

In summary this paper represents a valuable and important step in the progress of approaches to the non-invasive volumetric analysis - the "virtual unfolding" - of written material. I recommend it be accepted for publication with minor revisions that address my critique in this review.

Reviewer #2 (Remarks to the Author):

General Comment:

The submitted manuscript deals with an exciting topic which has attracted increasing attention during the last decade, the nondestructive access to hidden texts or other information in and on various kinds of media. While historic information on scrolls of metal foils, of parchment (e.g. the Dead Sea scrolls) and in a few cases of papyrus has been recovered and reported on in various publications, the authors of the submitted work introduce a different kind of stock of documents which has not yet gained much attraction, letters found in many trunks in various archives and collections, letters having many folds and having been often secured against opening by letterlocking. The work presented with this manuscript seems to show a major step forward in processing the "opening" of such letters, which are in relatively good preservation states, by algorithms without being influenced by a user (e.g. defining contour lines or similar). The achievement reported will certainly help in getting access to hidden text in cases of other substrate material as well, not as "homogeneous" as in the case of the letters dealt with here like from the Brienne collection. It will help to minimize user's influence in the process of automatic text recovery.

The procedure is convincingly described and the evidence presented in the manuscript appears to be solid. I am not able to claim, without any doubt, that the authors have really succeeded without any influence by an IT user in the algorithmic procedure. Maybe the co-reviewer(s) make a statement concerning this point. The presentation of the results justifies publication anyway.

A few general remarks and suggestions for improving the manuscript:

- Close to the beginning, in lines 51 to 53, three unopened letters are mentioned of different categories dealt with in this manuscript. Therefore, I suggest to reduce Figure 4 rigorously to the categories needed to illustrate the types of letters dealt with in this study. The full format chart should then be included in the Supplementary Materials part.
- Within the context of virtual unfolding, authors should be aware of a recent publication on unfolding of Egyptian papyri from the island of Elephantine (Louvre collection) with more than one fold (Virtual unfolding of folded papyri, Mahnke et al., Journal of Cultural Heritage 41 (2020)264). Maybe to be added in line 66.
- There is very little to nothing said about the substrate material. Presumably, the material of the Brienne letters presented here are in relatively good condition, quite homogeneous, so that the automatic algorithm approach works reasonably well. In case of large damage and in case of inhomogeneous material like papyrus, although structured but not regularly, such approach may need user-assisted unfolding.
- There is also little said about the ink (line 81). It is not necessary that the ink contains "metal" or metal ions. What is needed are high-Z elements for better contrast in the photo absorption, any nonmetallic compound as pigment (e.g. arsenic compound for reddish colored ink) would be good, and arsenic is no metal. And vice versa: magnesium is a metal, but a magnesium compound would not be good enough in producing contrast (there is no magnesium ink known! Just as an academic example).

Further detailed recommendations for improvements:

a) It seems to me unusual to give references in the abstract (see line 8, line 14, line 17).

b) line 25: one technology (in singular!).

c) line 34: The numbering of the figures should be changed. The first one mentioned should be Fig 1. I would not mention the figure 4 at this point, the first figure should be the one which is already labeled fig 1 presently (line 86). Fig. 4 as mentioned in line 170 should perhaps be reduced to the 3 categories used in this main part of the manuscript, while the full list should go into the supplement (as proposed already above under "General ..").

d) line 37: I would add, following "opened", "by the recipient".

e) line 49: "a fully automatic computational method" (singular).

f) line 52 to 57: It should be rephrased to get the logic proper, the present version starts with mentioning "three" prior to the description of the full Brienne collection out of which the three were chosen. I recommend, that somehow the first part of the sentence in line 56 should precede the line 52, the exact proper changes in terms of linguistics needed should be done by the authors.

g) line 86: figure 1: Perhaps in the caption, it should be explained that the enlarged parts B, D, and F lie in a different cross-sectional cut than the parts A, C, and E?

h) line 106: the symbols preceding the references 3 and 2 appear as the symbol "square" in my download.

i) line 119: the construction of the sentence is erroneous(?), maybe a missing comma or relative "that" or similar?

j) line 135: figure 2, part C: C is difficult to "read" in the figure. Is there a possibility to get a better contrast for the reconstruction? Different color?

k) line 169 to line 192: reducing to the categories needed for this main part of the manuscript needs adjustment in the text to properly refer to the supplement as the consequence.

In summary:

I highly recommend acceptance with the editorial adjustments suggested.

Reviewer #3 (Remarks to the Author):

Over the last few years numerous publications have come out on virtual unfoldings of complexly folded items/object, such as metal sheets as well as ancient papyri. This field is in rapid development, not least due to the technical developments within the fields needed in order to push such research forward. Most often such research deals with ancient items, since these are fragile and cannot be "opened" or unfolded without being damaged or destroyed. This creates issues in particular concerning the protection and conservation of cultural heritage in general. Therefore it is of utmost importance to make available new research, which deals with methods of unfolding efficiently, clearly and non-invasively, in order to let researchers who deal with the primary objects have access to new ways of approaching these items. Such items are often documents, including letters, spells and book manuscripts - all of which give immense amounts of detailed information about their time, the societies in which they were produced as well as the networks of which these items were part - both knowledge networks and transportation networks, as well as social, religious and intellectual networks. Therefore the manuscript under consideration comes as a welcome contribution to the wide variety of publications already available. I am commenting on the manuscript coming from the humanities and not as a computational expert. Therefore my observations do not have to do with the algorithm, but more broadly to do with the usefulness of the unfolded examples and the implications that these have for future studies.

The manuscript presents new virtual unfoldings of four letterpacks from the renaissance. These are complexly folded and have never been physically unfolded. The main intention with the submitted manuscript is to show how letterlocking in fact was applied on material in the past, which had to be transported between sender and receiver and treated in confidence. However, the method presented can of course also be applied to other kinds of complexly folded material, which never was intended to be unfolded again, such as magical amulets, spells etc (re. several of the references in the submitted manuscript).

The manuscript is very clear in its structure and explains the reader (both specialist and broader academic community) the process of the unfolding. Manuscript and the supplementary material held together presents an impressive array of material and instructively shows in which ways the algorithms applied in fact manages to unfold various kinds of letterlocked documents and just as importantly allows for the reading of the sheets in these letters. While the nature of this publication of course cannot be the content of these letters, this reviewer strongly suggests that another publication is prepared with the philological and historical implications found in these until now not read letters. I am sure that this has been considered already though. If a publication is underway already, a reference to such forthcoming study would be appreciated. A reference to this sort of publication approach could be added by citing the philological treatment of the Jerash Silver Scroll published after the Sci Rep. publication in 2015. Link: <https://journals.openedition.org/syria/4656>

I also recommend adding a reference to the Open Data of the in the submitted manuscript cited work on the Jerash Silver Scroll. Recently both the raw data pertaining to the first unfolding published in 2015 (Sci Rep. - cited in the submitted manuscript) and also new raw data relating to a

recent second unfolding (under publication) has been made available on figshare for free use by any interested parties as long as the work is acknowledged (data owners to be cited: Achim Lichtenberger and Rubina Raja). Link:

https://figshare.com/articles/Jerash_Silver_Scroll_Computed_Tomography_Data/12136380/1

as well as: <https://doi.org/10.6084/m9.figshare.12136380.v1>

If possible it would also be recommendable to make all raw data relating to the submitted article available as Open Data in connection with the publication of this article. This would support an Open Data policy and underline the willingness of the research team to share data so that other teams might reproduce results and potentially optimise methods in the future.

I support the publication of this manuscript in full.

AUTHOR'S RESPONSE TO REVIEWER 1:

While the concept of the virtual analysis of written material is not novel ("virtual unwrapping", "virtual unrolling", "virtual unfolding" - I prefer unwrapping even if folding or rolling was involved), some of the techniques that the authors have originated to meet the challenge are new and effective. In particular, some of the optimization methods used for refining the localization of surfaces within the volume, extending their reach within the volume to improve coverage and remove holes, pruning outliers, and measuring error in developability for the ultimate step of enforcing an isometry to a planar output are interesting and valuable.

In summary this paper represents a valuable and important step in the progress of approaches to the non-invasive volumetric analysis - the "virtual unfolding" - of written material. I recommend it be accepted for publication with minor revisions that address my critique in this review.

I think the authors may be inadvertently minimizing the complexity of the objects others have approached with this general framework when they say in the abstract and then the introduction "but these methods have been limited to documents with simple folding or rolling." The fact is that the letters in this paper appear quite healthy from a conservation standpoint. The paper is not delaminated, burned, stuck together, crushed, soaked to the point of fibrous deterioration, etc. Some might argue (I am not) that the problem of virtually opening these letters can be reduced to a set of linear layers that are connected through folds that, once identified, can allow linear sections to be re-assembled without the need for extra fretting over folds, interlocking sections, etc.

We thank the reviewer for these comments. By “simple folds” we were not referring to the simplicity of the prior work, but rather, a mathematical definition of a subset of folding: “A simple fold is a rigid rotation of some layers of paper around a single axis, which is the folding line” (source: <https://src.acm.org/binaries/content/assets/src/2013/hugoakitaya.pdf>). Since this language is quite domain-specific, we’ve removed this terminology to avoid confusion.

The Abstract now reads:

“Computational flattening algorithms have been successfully applied to X-ray microtomography scans of damaged historical documents, but have so far been limited to scrolls, books, and documents with one or two folds. The challenge tackled here is to reconstruct the intricate folds, tucks, and slits of unopened letters secured shut with “letterlocking,” a practice — systematized here for the first time — which underpinned global communications security for centuries before modern envelopes. We present a fully automatic computational pipeline for reconstructing and virtually unfolding volumetric scans of locked letters with complex internal folding, producing legible images of the letter’s contents and crease pattern while preserving letterlocking evidence....”

We have also added a section in the Discussion to address damage in the context of prior work:

“Our virtual unfolding pipeline builds on prior work using XMT to extract text hidden inside sealed, damaged, or otherwise unopenable historical documents [Applbaum1994]. Recent investigations lay out a

computational framework for virtually unwrapping or unrolling these documents to a flat isometry so their contents can be more readily observed; to date these investigations have focused primarily on scrolls [Seales2004, Lin2005, Mills2012; Neuber2012; Samko2014; Allegra2015; Barfod2015; Bukreeva2016; Seales2016; Liu2018; Rosin2018, Lichtenberge2020], books [Albertin2017, Stromer2017, Stromer2018], and artifacts folded once or twice [Albertin2017; Baum2017; Mahnke2020]. The primary motivation for many prior studies has been to recover text from highly damaged documents that cannot be physically opened and read. Damaged artifacts and highly fibrous, delaminated, or otherwise heterogeneous writing substrates pose particular problems for segmentation, and severe damage-induced warping may be difficult to virtually flatten without introducing heavy distortion of the surface. Because the artifacts we study are made from a largely homogenous paper material that is relatively undamaged, our focus instead has been to fully automate and extend existing computational flattening frameworks to tackle a new class of documents with more intricate folding than has previously been explored.”

...

“A potential obstacle to applying this pipeline to highly warped or damaged documents is our incremental flattening approach, which maps the results of segmentation to \mathbb{R}^2 . Because our datasets have so far been limited to artifacts with little warping, we do not have to introduce much distortion into the mesh to compute this mapping (Fig. 8), and our incremental approach is usually sufficient. Many historical artifacts are warped due to excessive heating, water damage, delamination, decay, or other processes of time [Pal2014], and may fail to map properly to 2D in our current scheme....”

I would like to point out that the authors should consider citing as well the work on manuscript fragments within book bindings, which was the first application similar to this work on letters: the process was defined, tomography was the basis, and the material was not morphological like cuneiform or inscribed surfaces but rather was ink-based:

W. B. Seales, Y. Lin, Digital restoration using volumetric scanning, in Proceedings of the 2004 Joint ACM/IEEE Conference on Digital Libraries (IEEE, 2004)

Y. Lin, Physically-Based Digital Restoration Using Volumetric Scanning (University of Kentucky, Lexington, 2007) Ph.D. thesis.

Y. Lin, W. B. Seales, Opaque document imaging: Building images of inaccessible texts, in Tenth IEEE International Conference on Computer Vision (ICCV'05) (IEEE, 2005)

We agree with the reviewer’s comment and have added references to Seales2004 and Lin2005 in our discussion of prior work. We were not able to access an online version of Lin2007 Doctoral Thesis (we have submitted an ILL request for a digital copy), so we were not able to include this reference in the current draft.

I would simply caution against calling rolled material, or layers within a book or codex, as "simple." Most of it that I have seen, because of damage, is riddled with uncertainty as to what is folded, broken, crushed, full of holes, bubbled because of burns, etc. So perhaps making it clear that your challenge is shape / category of interlocking letter, not damage. Others deal with damage, and it has its own challenges.

I would like to point out also that the substrate of paper you are viewing in this problem - which is not burned or delaminated or apparently damaged in any way - is really the best case for segmentation.

Likewise, for texturing, the ink gives a nice density-based signal, obviating the need to do any processing at the texturing step other than to situate what is in the volume onto the segmented surfaces in an accurate way. That is fine for this problem - just making the point that it is possible with different materials to face more complicated issues at that step in the processing.

Also I appreciate the desire to highlight the part of this work that should be seen as novel, but I would caution against overreach with which part of these methods are new versus which parts you have appropriated - perhaps in a different/novel way for the particulars of this kind of letter/material. General virtual unwrapping has been postulated for a while (based on any kind of volumetric imaging) followed by software steps (in abstraction, such as "segmentation" and "texturing" and "unwrapping". Your work fits nicely as a variation into those already-articulated frameworks, and you could actually make that alignment if you chose to...to argue that you are refining / innovating a new "unfolding" step in the generally accepted pipeline of virtual analysis of written material.

We agree with this feedback from the reviewer and have modified this section (now moved to the Discussion) to situate our work within previous studies in the way that the reviewer describes. These changes are also reflected in the modified Abstract.

Another point I believe could be made more crisp is the reality of physical opening. Is it not possible to open these letters at all through paper conservation? Would doing that ruin something? I realize that when the seal is intact, it would need to be broken - but my understanding from the paper was that the ones that were scanned did not have seals (because the lead would cause artifacts in the scan). So how do you argue against someone who might say: "keep a few exemplars intact, but open the others as a physical conservation project for the purpose of better imaging of the text." The argument I have used is that in some cases physical restoration is impossible, e.g., it will destroy the object. But I don't think you can say that here: which part of this letter object is destroyed if you open it?

We thank the review for this feedback. Letters in the collection that did not use adhesive at all – in other words that only used “fold” and “tuck” letterlocking manipulations – were indeed opened and reclosed under careful scrutiny as part of our larger exploration of the Brienne archive. The letters investigated here were sealed shut with adhesive, but that adhesive did not contain lead elements. In this historical period, some adhesive was made of sealing wax, which often contains lead, although not always; other adhesives included clear starch wafers, which do not create beam scatter. The letters chosen here were preselected for having been secured with methods that did not employ lead-containing elements. These letters could have been opened using traditional methods, i.e. cutting around the seal, but this would destroy the integrity of the packet as an intact unopened artifact. We have tried to present a clear argument about this decision in the main article by rephrasing the following sentence: “Until now, our analysis of letterlocking has been limited by the standard archival practice of cutting open sealed letters on request, compromising the physical integrity of the unopened letterpacket.”

You do make the statement in talking about prior work as distinguished from yours that "nearly all methods are 2.5 D (and not fully 3D)." True that many have approached the problem from a slice-based position, but not everyone. Our own work uses a 3D structure tensor in the volume and other fully-3D approaches - I am sure we are not the only ones.

We thank the reviewer for this comment. By 2.5D we were referring to the fact that even though some methods do use a 3D structure tensor or 3D filtering kernels, most of the prior work makes a strong assumption about similar layer to layer topology of slices across the length of the scan when constructing a mesh (often resulting in a very regular, gridded mesh structure). These are assumptions we did not make in our methods as they would not hold for the folded artifacts studied here. We have tried to clarify this in the Discussion:

“...Each letterpacket was constructed with a distinct folding sequence: some packets contain inner folds that are angled diagonally relative to their outer silhouette; others are bound together by a separate paper lock. Rather than linking together results from many independent cross-sectional analyses of the volumetric data (assuming some degree of global layer congruence across the length of the scan [Seales2004, Lin2005, Neuber2012, Samko2014, Allegra2015, Barfod2015, Seales2016, Liu2018, Rosin2018, Baum2017; Mahnke2020]), we utilize a fully 3D geometric analysis that is unbiased to fold orientation and makes no assumptions about the folded topology of the packet....”

*It wasn't straightforward for me to understand how the unfolding algorithm detects automatically where the folds are...it seems like the mapping from e_3 to e_2 has to happen piecewise for each of many "connected components", and then those individual components, after "mesh propagation", end up being put into a common frame by solving for a rotation/translation, and then whatever *that* turns out to be helps you infer that a "fold" or "hinge" is happening...But it is hard to tell if I'm right about this - if anything is ever explicitly identified as a fold or a crease or crack, or how that stage works in the end. [aside: I see that you are nudging the texture with curvature info to build a visual cue around "crease", but that's not what I mean]. Maybe a high level walk-through first could help the discussion.*

We agree with the organizational changes suggested and have reorganized the Virtual Unfolding to give a high-level walkthrough of the pipeline first, then we discuss comparisons to previous work in the Discussion. We have added more specific information about the algorithm steps in the Virtual Unfolding section, including some clarifying statements about the connected components, how the connected components are merged, and how crease patterns (images) are generated.

Currently the crease pattern we generate is a pixelated image mapping curvature to pixel color. We do not do any further analysis of this image to try to pull out salient features and construct a more minimal (eg vectorized) representation of the crease pattern. The resulting crease pattern image is analyzed by a human to make inferences about the location and function of folds. We have included a zoomed in view of the crease pattern in Figure 2 to help make this more clear and included the following text in the “Virtual Unfolding” section:

“A similar process is used to generate an image of the crease pattern; in this case we map the mean curvature of the 3D mesh to 2D (Fig. 4: K, L). The sign of the curvature corresponds to the direction (mountain or valley) of the crease, indicated by red or blue, respectively. Color saturation specifies the magnitude of curvature, with white indicating zero curvature. The resulting crease pattern images show sharper creases as thin lines of saturated color, and more gentle creases as broad, faded lines. In addition

to the folding pattern, other features that introduce curvature into the writing substrate — such as buckling near the corners of a letterpacket — are made visible (Fig. 2: F). In combination with the 3D reconstructions, these crease patterns enable researchers to recreate the step-by-step locking process that would be needed to turn a flat sheet of paper into the finished packet (Fig. 1: D).”

The figure 2 really has too much in it (and it is too tiny) and I think it would be helpful to expand that if there is space.

We agree with the reviewer's comment and have expanded Figure 2 into three figures (Figures 2, 4, 5). In the new figures we have enlarged some images so they are more clearly visible. We have also included some additional images to help clarify the computational pipeline.

You claim that these segmentation, unfolding, and texturing methods are fully automatic ("Virtual Unfolding", page 2). That is important and I think can and should be made clearer (and more believable) if you focus on explaining it a little better. My experience is that "fully automatic" is always defeated by the condition of certain items (pages too close together; not enough spatial resolution in the volume; ink variations; etc.) It might be instructive for you to show examples where your method fails in order to give the reader insight into the failure modes and the visual consequence in the final unfolded image of full automation when things go wrong.

In the end I am left wondering about how much overall distortion there is in the flattened and unfolded letter pages. Since they are undamaged and folding is non-elastic, they ought to map quite readily isomorphically to a plane. I would guess the layers are all essentially flat, meaning there isn't much digital flattening that needs to happen during the mesh optimization. Is that true? Would it be possible to show distortion maps / heat maps from the 3D segmentation to the 2D flattened image to know where there was page distortion (geometric) from the digital unfolding step? I don't think I mean a "distance map" here, but a visual measure of the distortion on a section that had to occur to go from E3 to E2.

We agree with the reviewer's comment and have included the distance map from E3 to E2 in Figure 8. We have added more specifics about the automated pipeline in the Virtual Unfolding section and discuss its limitations in the Discussion:

“To facilitate widespread adoption of virtual unfolding, future work could include improving robustness. Our methods rely heavily on local interactions between vertices in the mesh; while efficient to compute, these local rules can be prone to global consistency errors. This is exacerbated by the presence of scanning artefacts (Fig. 3), especially those caused by leaded seals (i.e. those containing red lead oxide) or shellac seals containing strands of metal, which were avoided in this study (the seal shown in Fig. 3 F is unleaded). Errors in segmentation and flattening may accumulate into localized distortions of the 2D embedding and result in incomplete or incorrect merging of connected components during hybrid mesh propagation; the missing pieces of letters DB-1538 and DB-1627 are a result of these merging issues (Fig. 8). Future work may include additional passes to fill in the gaps in the reconstruction or user-in-the-loop mechanisms to resolve segmentation errors early in the process....“

In the supplementary description the domain experts compare the image to the quality of micro-film. I checked a few numbers: at 38 micron voxel resolution you should be able to get somewhere around 660dpi on the final image (subject to stretching, etc.: this is just $v / \text{"voxel's resolution in microns"}$). So you could get 1200dpi if you scanned at 20 microns. So that's one point - of course, maybe that isn't possible with the kit you used. A related point is that a 40MP digital camera will give you 1000-1200 dpi on a letter sized page, and that is considered archival quality. So if you could get just a bit better at the scan stage you would be there with your output final textures. I wonder if your domain experts would find that to be sharper, more like Photography?

We thank the review for this comment and have now included information about DPI in the Virtual Unfolding section: “At the current scanning resolution, our texturing results are comparable to a 668dpi digital image.”

This paragraph was originally written from the perspectives of historians on the project who have spent years reading material on microfilm, and was meant to give a general impression of the ability of the images we produced to substitute for seeing the substrate surface itself, in terms of legibility to the kinds of readers likely to access it. You are right that such a comparison is too general in this context, without reference to specifics, and we have removed the reference to microfilm at this point in the paper, reflecting instead (at the urging of Reviewer 3) on the impact we anticipate this work will have on historical studies.

Final thought is one of provenance chains. By that I mean your final texture/image claims are derived directly from the original scan through a series of complex algorithmic operations you define. People working in this field should consider that extracted information from the interior of artifacts that can never be opened for inspection are hard to subject to peer scrutiny. There is no way as a reviewer of your work that I can truly verify your final texture claims - in particular, the textual implications of what you claim to have revealed from within these letters. You might acknowledge that somehow, and suggest some way to build confidence in those of us who must trust that the revealed text is authentically supported by the physical object and the volumetric scan you made. I consider it an open problem as yet, but if I grabbed your volume and pushed data the other way through the pipeline we could create/change text rather than reveal it...creating a pathway for inscrutable results that will not help the cause of non-invasive analysis.

We thank the reviewer for this comment. We do include a validation study using an opened and refolded letter (DB-2040), and compare our results to photographs of the letter's contents - but we agree with you that we do not have a way to verify the results for the unopened letters studied here. Additionally, we have included more information about limitations to the technique in the Discussion. We hope this addresses the reviewer's concern.

AUTHOR'S RESPONSE TO REVIEWER 2:

The work presented with this manuscript seems to show a major step forward in processing the "opening" of such letters, which are in relatively good preservation states, by algorithms without being influenced by a user (e.g. defining contour lines or similar). The achievement reported will certainly help in getting access to hidden text in cases of other substrate material as well, not as "homogeneous" as in the case of the letters dealt with here like from the Brienne collection. It will help to minimize user's influence in the process of automatic text recovery. The procedure is convincingly described and the evidence presented in the manuscript appears to be solid. I am not able to claim, without any doubt, that the authors have really succeeded without any influence by an IT user in the algorithmic procedure. Maybe the co-reviewer(s) make a statement concerning this point. I highly recommend acceptance with the editorial adjustments suggested.

We thank the reviewer for these comments. In response to the full-automation claimed in the paper: we have included source code, raw data, and animations of the virtual unfolding process on our supplementary website brienne.org/unfolding (password unfolding). This website also provides animations of the flattening and hybrid mesh propagation proceeding for several example files of full-sized letters. The README in the source code gives instructions for processing a small sample file on a regular computer (e.g. a laptop). The full-sized files require the use of a large GPU with 16GB RAM.

We do acknowledge that the process is governed by several parameters, including some that dictate the expected thickness of the writing substrate and additional parameters for tuning the optimization process. We were able to complete analysis of all four letters described in the paper using the same parameters. The location of these parameters is discussed in the README.

We will be publishing all code to Github when the paper is published.

A few general remarks and suggestions for improving the manuscript: - Close to the beginning, in lines 51 to 53, three unopened letters are mentioned of different categories dealt with in this manuscript. Therefore, I suggest to reduce Figure 4 rigorously to the categories needed to illustrate the types of letters dealt with in this study. The full format chart should then be included in the Supplementary Materials part.

Thank you for this suggestion, which makes sense. In order to address this query, we have introduced, as part of a new Figure 1, an image of the letterlocking category for the specific letter being illustrated (Fig. 1: E). Figure 1 now progresses as follows: an image of the letterpacket under investigation, front and back; an x-ray of that packet to show its multiple layers while closed; a diagram of its assigned crease pattern in its unfolded state; a step-by-step folding sequence to show how a flat substrate can be turned into that packet; and then, based on that reconstruction, the letterlocking category assigned to it. We strongly believe that it is important for the whole categories and formats chart (Fig. 7) to be part of the main article, since this study supports the creation of that chart as one of the key research findings; this is the first ever publication of this chart which supports the launch of letterlocking as a new field of study, so it needs to be prominent in the main article to make sense of the categories of our case studies.

- Within the context of virtual unfolding, authors should be aware of a recent publication on unfolding of Egyptian papyri from the island of Elephantine (Louvre collection) with more than one fold (Virtual unfolding of folded papyri, Mahnke et al., *Journal of Cultural Heritage* 41 (2020)264). Maybe to be added in line 66.

We thank the review for this comment and have added this reference in our discussion of prior work.

- There is very little to nothing said about the substrate material. Presumably, the material of the Brienne letters presented here are in relatively good condition, quite homogeneous, so that the automatic algorithm approach works reasonably well. In case of large damage and in case of inhomogeneous material like papyrus, although structured but not regularly, such approach may need user-assisted unfolding.

We agree with the reviewer's comments. In response, we have added a section to address this in the Discussion:

"...The primary motivation for many prior studies has been to recover text from highly damaged documents that cannot be physically opened and read. Damaged artifacts and highly fibrous, delaminated, or otherwise heterogeneous writing substrates pose particular problems for segmentation, and severe damage-induced warping may be difficult to virtually flatten without introducing heavy distortion of the surface. Because the artifacts we study are made from a largely homogenous paper material that is relatively undamaged, our focus instead has been to fully automate and extend existing computational flattening frameworks to tackle a new class of documents with more intricate folding than has previously been explored...."

Later on in the Discussion we have addressed some limitations of our pipeline with regard to damage (specifically severe warping of the substrate):

"A potential obstacle to applying this pipeline to highly warped or damaged documents is our incremental flattening approach, which maps the results of segmentation to \mathbb{R}^2 . Because our datasets have so far been limited to artifacts with little warping, we do not have to introduce much distortion into the mesh to compute this mapping (Fig. 8), and our incremental approach is usually sufficient. Many historical artifacts are warped due to excessive heating, water damage, delamination, decay, or other processes of time [Pal2014], and may fail to map properly to 2D in our current scheme...."

- There is also little said about the ink (line 81). It is not necessary that the ink contains "metal" or metal ions. What is needed are high-Z elements for better contrast in the photo absorption, any nonmetallic compound as pigment (e.g. arsenic compound for reddish colored ink) would be good, and arsenic is no metal. And vice versa: magnesium is a metal, but a magnesium compound would not be good enough in producing contrast (there is no magnesium ink known! Just as an academic example).

We agree with the reviewer's comment and have modified the description of the scanning to include the line:

“XMT scanning produces a volumetric dataset representing material density in 3D space; inks containing higher Z elements (such as iron, copper, and mercury) result in bright (high-density) regions within the scan (Fig. 3).”

Further detailed recommendations for improvements:

a) *It seems to me unusual to give references in the abstract (see line 8, line 14, line 17).*

We agree with the reviewer’s comment and have modified the abstract to fit within the guidelines for submission to this publication (including removing these references and shortening it to the correct word count limit).

b) *line 25: one technology (in singular!).*

We thank the reviewer for this comment. Grammatically, we couldn’t make this work to fit the current claim: “The letter is one of the most important communication technologies in human history.” We don’t think we could substantiate the claim that the letter is *the* most important technology, period. We have therefore stuck with technologies plural, but are happy to change if we have misunderstood what was being asked.

c) *line 34: The numbering of the figures should be changed. The first one mentioned should be Fig 1. I would not mention the figure 4 at this point, the first figure should be the one which is already labeled fig 1 presently (line 86). Fig. 4 as mentioned in line 170 should perhaps be reduced to the 3 categories used in this main part of the manuscript, while the full list should go into the supplement (as proposed already above under "General ..").*

We agree with the reviewer’s comment and have referenced the figures in the order that they appear in the article. The reference to Figure 4 has been removed from the introduction.

d) *line 37: I would add, following "opened", "by the recipient".*

We thank the reviewer for this comment and have changed the sentence to: “...since letters are designed to be opened at their intended destination...”. Historians on our team point out that in early modern Europe, the notion of a single “recipient” is complicated, since many people used secretaries or family members to read the letter before handing it on.

e) *line 49: "a fully automatic computational method" (singular).*

We have fixed the typo so that it now reads “a fully automatic computational method”.

f) *line 52 to 57: It should be rephrased to get the logic proper, the present version starts with mentioning "three" prior to the description of the full Brienne collection out of which the three were chosen. I recommend, that somehow the first part of the sentence in line 56 should precede the line 52, the exact proper changes in terms of linguistics needed should be done by the authors.*

We agree with the reviewer's comment. Due to changes we have made in response to another reviewer (reviewer 1), we have restructured this section so that the Virtual Unfolding section focuses on a high-level description of the method, and have moved the description of the specific letterpackets studied to the results and discussion sections. We believe these changes satisfy this request.

g) line 86: figure 1: Perhaps in the caption, it should be explained that the enlarged parts B, D, and F lie in a different cross-sectional cut than the parts A, C, and E?

We agree with the reviewer's comment and have adjusted the figure (renamed to Figure 3) to show two cross sections, from which all the enlarged images are pulled.

h) line 106: the symbols preceding the references 3 and 2 appear as the symbol "square" in my download.

This is in reference to the symbols for \mathbb{R}^2 and \mathbb{R}^3 (\mathbb{R}^2 and \mathbb{R}^3). We will double check this when submitting, but hope that the journal's computers will be able to process this symbol.

i) line 119: the construction of the sentence is erroneous(?), maybe a missing comma or relative "that" or similar?

We have added the word "that" to this sentence.

j) line 135: figure 2, part C: C is difficult to "read" in the figure. Is there a possibility to get a better contrast for the reconstruction? Different color?

We acknowledge that this image is difficult to read, unfortunately some of the problem is due to double-sided text and poor ink contrast which is not solvable by changing the color/contrast of the image or offsetting texturing along the surface normal of the mesh. We have adjusted Figure 2 so that the images are larger and have included a zoomed in view to more clearly show the text. We have added a reference to this figure in the discussion:

"Close inspection of the results for DB-1538 (Fig. 2: C, E) show that ink contrast and legibility could also be improved, possibly by using alternative scanning methods [Mocella2015] or by performing additional post-processing on the texturing results."

k) line 169 to line 192: reducing to the categories needed for this main part of the manuscript needs adjustment in the text to properly refer to the supplement as the consequence.

Please see our comments above about the importance of the presence of Figure 7 in the main body of the article.

AUTHOR'S RESPONSE TO REVIEWER 3:

Over the last few years numerous publications have out come on virtual unfoldings of complexly folded items/object, such as metal sheets as well as ancient papyri. This field is in rapid development, not least due to the technical developments within the fields needed in order to push such research forward. Most often such research deals with ancient items, since these are fragile and cannot be "opened" or unfolded without being damaged or destroyed. This creates issues in particular concerning the protection and conservation of cultural heritage in general. Therefore it is of utmost importance to make available new research, which deals with methods of unfolding efficiently, clearly and non-invasively, in order to let researchers who deal with the primary objects have access to new ways of approaching these items. Such items are often documents, including letters, spells and book manuscripts - all of which give immense amounts of detailed information about their time, the societies in which they were produced as well as the networks of which these items were part - both knowledge networks and transportation networks, as well as social, religious and intellectual networks. Therefore the manuscript under consideration comes as a welcome contribution to the wide variety of publications already available. I am commenting on the manuscript coming from the humanities and not as a computational expert. Therefore my observations do not have to do with the algorithm, but more broadly to do with the usefulness of the unfolded examples and the implications that these have for future studies. I support the publication of this manuscript in full.

The manuscript presents new virtual unfoldings of four letterpackets from the renaissance. These are complexly folded and have never been physically unfolded. The main intention with the submitted manuscript is to show how letterlocking in fact was applied on material in the past, which had to be transported between sender and receiver and treated in confidence. However, the method presented can of course also be applied to other kinds of complexly folded material, which never was intended to be unfolded again, such as magical amulets, spells etc (re. several of the references in the submitted manuscript).

The manuscript is very clear in its structure and explains the reader (both specialist and broader academic community) the process of the unfolding. Manuscript and the supplementary material held together presents an impressive array of material and instructively shows in which ways the algorithms applied in fact manages to unfold various kinds of letterlocked documents and just as importantly allows for the reading of the sheets in these letters. While the nature of this publication of course cannot be the content of these letters, this reviewer strongly suggests that another publication is prepared with the philological and historical implications found in these until now not read letters. I am sure that this has been considered already though. If a publication is underway already, a reference to such forthcoming study would be appreciated. A reference to this sort of publication approach could be added by citing the philological treatment of the Jerash Silver Scroll published after the Sci Rep. publication in 2015. Link: <https://journals.openedition.org/syria/4656>

We thank Reviewer 3 very much for these very useful comments, and particularly welcome the opportunity to make more of a case for the potential impact of our work on historical and cultural studies. These are highlighted in blue in the final pages.

With reference to the Jerash Silver Scrolls project, we're grateful to read of a project that has been organised in a similar way to ours, in terms of publishing scientific discoveries first, then cultural history, and curating data online, though we didn't think the publishing approach itself warranted a citation. However, we are very pleased to acknowledge this important work and in fact, have added further reference to it (see response to next comment).

I also recommend adding a reference to the Open Data of the in the submitted manuscript cited work on the Jerash Silver Scroll. Recently both the raw data pertaining to the first unfolding published in 2015 (Sci Rep. - cited in the submitted manuscript) and also new raw data relating to a recent second unfolding (under publication) has been made available on figshare for free use by any interested parties as long as the work is acknowledged (data owners to be cited: Achim Lichtenberger and Rubina Raja). Link: https://figshare.com/articles/Jerash_Silver_Scroll_Computed_Tomography_Data/12136380/1 as well as: <https://doi.org/10.6084/m9.figshare.12136380.v1>

We thank the reviewer for these comments and have added a reference to the Open Data for the Jerash Silver Scroll in our discussion of prior work.

If possible it would also be recommendable to make all raw data relating to the submitted article available as Open Data in connection with the publication of this article. This would support an Open Data policy and underline the willingness of the research team to share data so that other teams might reproduce results and potentially optimise methods in the future.

Yes, we agree with the reviewer and we plan to release the raw and result data (about 50GB) with the publication. We are currently hosting all data (including the original scan of 10 letters, and individual letters cropped from this dataset), source code, and results on our supplementary website brienne.org/unfolding. This website will be made public when the article is published. The source code will be published on Github.

REVIEWER COMMENTS

Reviewer #1 (Remarks to the Author):

You have adequately addressed most points from my review in the first round.

There are two points where I think you should still consider a clarification in this manuscript.

The first is for you to consider specifically remarking on the importance of you and other researchers providing for some way to follow the data provenance from the scanned volume to the final "texture claim."

Although you may not have provided a framework right now for outside reviewers / researchers / scholars to visualize this pathway, the integrity of the processing pipeline from volume to texture is crucial in cases where the original physical object will never be opened.

It may come to the point where even the scanning as a system will need to provide a certified imprint on its data to confirm that the scan was of the true physical object.

But the chain of evidence from the scan through the steps you outline to the final claim are crucial as a "data provenance" question, and the integrity of it matters for the scholarship that will happen based on the final results.

Note that this is a somewhat higher bar than just providing all the data on a website.

I encourage you to take a look at this paper:

<http://cidrdb.org/cidr2020/papers/p8-agrawal-cidr20.pdf>

and review their take on provenance chains surrounding the data in machine learning environments.

The problems are similar, where scholars use results from a complex data transformation pipeline without fully understanding or being able to reproduce those stages.

The second point is to say just a bit more about how crucial it is that your inks have the "higher Z elements" in order for your method to produce readable text.

Clearly all the geometry of the folding is captured in the volumetric scan independently of

the ink chemistry, but the ink signature is only captured by your method if the ink has a discernible density signature from your texturing method.

There are other things that can be done when the ink does not fit this profile, as in this work:

<https://journals.plos.org/plosone/article?id=10.1371/journal.pone.0215775>

But if you scan a closed letter with your current approach and it was written with a charcoal pencil or painted in watercolor you would only be able to reveal structure, not writing

(if the ink evidence did not show up as a clear and visible density shift in the scan). I have scanned objects like cartonnage that contain multiple ink types, some of which are very difficult to see in tomography. Objects like "books of the dead" contain a variety of pigments/inks, not all of which might show up equally well. One could imagine artificial gaps in a mixed-ink letter based on the chemistry of the inks.

So when you open a letter signed in blood, the signature will likely not appear in this method (unless someone was taking lots of iron supplements? or had heavy metal poisoning?)

After your consideration of these final points I recommend this paper for publication.

I wish you all the best in your continued pursuit of this work.

Reviewer #2 (Remarks to the Author):

As already stated in my previous report, the submitted manuscript deals with an exciting topic which has attracted increasing attention during the last decade, the nondestructive access to hidden texts or other information in and on various kinds of media. The revised manuscript has been improved considerably. The authors have addressed all points raised in my review point by point (as well as by the co-reviewers, as far as I can see from the response letter). Especially the improvements with the figures, their ordering, descriptions and captions should be mentioned.

Two minor points should be added and ask for correction:

- 1) In the list of references, there is one reference, ref. 31, where the list of authors is abbreviated by "et al.". The full list of authors should be given like in other references, even with a longer list of authors.
- 2) In the chapter "Discussion", page 17, line 248, 262 and 265, reference numbers are given from the list of references, the ordering within the brackets [...] seems to be arbitrarily chosen, if there is no special reason, one would expect an ordering from the lowest to the highest number.

In summary, I recommend to accept the manuscript for publication.

AUTHOR'S RESPONSE TO REVIEWER 1:

You have adequately addressed most points from my review in the first round.

There are two points where I think you should still consider a clarification in this manuscript.

The first is for you to consider specifically remarking on the importance of you and other researchers providing for some way to follow the data provenance from the scanned volume to the final "texture claim."

Although you may not have provided a framework right now for outside reviewers / researchers / scholars to visualize this pathway, the integrity of the processing pipeline from volume to texture is crucial in cases where the original physical object will never be opened.

It may come to the point where even the scanning as a system will need to provide a certified imprint on its data to confirm that the scan was of the true physical object.

But the chain of evidence from the scan through the steps you outline to the final claim are crucial as a "data provenance" question, and the integrity of it matters for the scholarship that will happen based on the final results.

Note that this is a somewhat higher bar than just providing all the data on a website.

I encourage you to take a look at this paper:

<http://cidrdb.org/cidr2020/papers/p8-agrawal-cidr20.pdf>

and review their take on provenance chains surrounding the data in machine learning environments.

The problems are similar, where scholars use results from a complex data transformation pipeline without fully understanding or being able to reproduce those stages.

We agree with the reviewer's comment and have added the following section to the Discussion:

"In general, all files created from our pipeline are derivations of the original scan data and code; given changes to either of these inputs, we would expect the results to change. As improvements are made to the pipeline, it is crucial to track metadata such as the code version and changes to input parameters to ensure reproducibility of results. Additionally, tools to visualize the transformation of data through the pipeline will give outside researchers a window into the data provenance, affirming the integrity of the results, especially in cases where the original physical artifact remains unopened."

We would also like to comment that we are actively taking steps towards addressing data provenance. We have updated the latest version of our code to save an output file containing the metadata from the scan (e.g. voltage, exposure time, Feldkamp back-projection parameters) as well as the parameters for the run (e.g. gaussian params, simulation params) and the git hash of the current state of the code. Additionally, we are currently building a web-based tool for visualizing each of the intermediate files produced during the processing pipeline so a user

can explore the pipeline at a more granular level. We are aiming to have this tool available upon publication.

The second point is to say just a bit more about how crucial it is that your inks have the "higher Z elements" in order for your method to produce readable text. Clearly all the geometry of the folding is captured in the volumetric scan independently of the ink chemistry, but the ink signature is only captured by your method if the ink has a discernible density signature from your texturing method.

There are other things that can be done when the ink does not fit this profile, as in this work: <https://journals.plos.org/plosone/article?id=10.1371/journal.pone.0215775>

But if you scan a closed letter with your current approach and it was written with a charcoal pencil or painted in watercolor you would only be able to reveal structure, not writing (if the ink evidence did not show up as a clear and visible density shift in the scan). I have scanned objects like cartonnage that contain multiple ink types, some of which are very difficult to see in tomography. Objects like "books of the dead" contain a variety of pigments/inks, not all of which might show up equally well. One could imagine artificial gaps in a mixed-ink letter based on the chemistry of the inks.

So when you open a letter signed in blood, the signature will likely not appear in this method (unless someone was taking lots of iron supplements? or had heavy metal poisoning?)

We agree with the reviewer's comment and have added the following to the Discussion to address this comment:

"Inks with a similar density as the underlying writing substrate (e.g. carbon-based inks) may have low contrast (Fig. 2: C, E) or no discernible signal at all (see DB-1976 result in Supplementary Materials). Ink contrast and legibility could be improved by using alternative scanning methods [32] or by performing additional post-processing on the unfolded results to amplify small changes in density or structure from the presence of ink [33]."

We have added <https://journals.plos.org/plosone/article?id=10.1371/journal.pone.0215775> as reference 33.

After your consideration of these final points I recommend this paper for publication. I wish you all the best in your continued pursuit of this work.

AUTHOR'S RESPONSE TO REVIEWER 2:

*As already stated in my previous report, the submitted manuscript deals with an exciting topic which has attracted increasing attention during the last decade, the nondestructive access to hidden texts or other information in and on various kinds of media. **The revised manuscript has been improved considerably. The authors have addressed all points raised in my review point by point (as well as by the co-reviewers, as far as I can see from the response letter).** Especially the improvements with the figures, their ordering, descriptions and captions should be mentioned.*

Two minor points should be added and ask for correction:

1) In the list of references, there is one reference, ref. 31, where the list of authors is abbreviated by "et al.". The full list of authors should be given like in other references, even with a longer list of authors.

We have fixed this citation.

2) In the chapter "Discussion", page 17, line 248, 262 and 265, reference numbers are given from the list of references, the ordering within the brackets [...] seems to be arbitrarily chosen, if there is no special reason, one would expect an ordering from the lowest to the highest number.

We have fixed the citation ordering.

In summary, I recommend to accept the manuscript for publication.

REVIEWERS' COMMENTS:

Reviewer #1 (Remarks to the Author):

The revisions to the manuscript in response to the comments I made in the last round (regarding data provenance and the inks having higher Z elements) are satisfactory.

I recommend publication.

And bravo for releasing the visualization tool for the provenance chain (and for thinking about how to store the parameters of the chain as metadata).